

# Fear of becoming pregnant among female healthcare students in Spain

Felipe Navarro-Cremades[1], Antonio Palazón-Bru[1],
María del Ángel Arroyo-Sebastián[2], Luis Gómez-Pérez[3], Armina Sepehri[1],
Salvador Martínez-Pérez[4], Dolores Marhuenda-Amorós[3],
María Mercedes Rizo-Baeza[5] and Vicente Francisco Gil-Guillén[1]

[1] Department of Clinical Medicine, Miguel Hernández University, San Juan de Alicante, Alicante, Spain
[2] Health Centre of San Juan de Alicante, Conselleria de Sanitat, San Juan de Alicante, Alicante, Spain
[3] Department of Pathology and Surgery, Miguel Hernandez University, San Juan de Alicante, Alicante, Spain
[4] Department of Histology and Anatomy, Miguel Hernandez University, San Juan de Alicante, Alicante, Spain
[5] Department of Nursing, University of Alicante, San Vicente del Raspeig, Alicante, Spain

Corresponding author
Antonio Palazón-Bru,
antonio.pb23@gmail.com

## ABSTRACT

The inconsistent use of hormonal contraceptive methods can result, during the first year of use, in one in twelve women still having an undesired pregnancy. This may lead to women experiencing fear of becoming pregnant (FBP). We have only found one study examining the proportion of FBP among women who used hormonal contraceptives. To gather further scientific evidence we undertook an observational, cross-sectional study involving 472 women at a Spanish university in 2005–2009. The inclusion criteria were having had vaginal intercourse with a man in the previous three months and usual use for contraception of a male condom or hormonal contraceptives, or no method of contraception. The outcome was FBP. The secondary variables were contraceptive method used (oral contraceptives; condom; none), desire to increase the frequency of sexual relations, frequency of sexual intercourse with the partner, the sexual partner not always able to ejaculate, desire to increase the partner's time before orgasm, age and being in a stable relationship. A multivariate logistic regression model was used to determine the associated factors. Of the 472 women, 171 experienced FBP (36.2%). Factors significantly associated ($p < 0.05$) with this FBP were method of contraception (condom and none), desire to increase the partner's ability to delay orgasm and higher frequency of sexual intercourse with the partner. There was a high proportion of FBP, depending on the use of efficient contraceptive methods. A possible solution to this problem may reside in educational programmes. Qualitative studies would be useful to design these programmes.

# INTRODUCTION

The last 35 years have seen significant advances in the development of contraceptive methods (*World Health Organization, 2010*). This has resulted in these methods being highly effective if used correctly, particularly the hormonal contraceptives (*World Health Organization, 2011*; *World Health Organization, 2010*). Nevertheless, a certain percentage

of women use hormonal contraceptive methods inconsistently, either due to side effects or to lack of compliance taking them (*Rosenberg, Waugh & Meehan, 1995*; *Lete et al., 2008*). During the first year of use of these methods approximately one in every 12 women may have an unwanted pregnancy (*Trusell & Kost, 1987*). For these reasons women, even if they use hormonal contraceptive methods, may experience fear of having an unwanted pregnancy. This is translated into relief when they start their monthly menstruation after having had vaginal intercourse with a man who ejaculated (*Lete et al., 2008*).

Considering the lack of papers examining this fear, we undertook a study in a population of university students studying the healthcare sciences, assessing which women experienced fear of becoming pregnant (FBP). Unlike other studies (*Lete et al., 2008*), our study also included women who commonly relied on a male condom for hormonal contraception and women who did not use any contraceptive method at all. We also determined the factors related with their sexual behaviour that could be associated with this FBP. The results suggest the need for more education about preventing an undesired pregnancy in future healthcare workers.

## MATERIALS & METHODS

### Study population

The study population comprised female students of the faculties teaching healthcare sciences (Medicine, Physiotherapy, Pharmacy, Podiatry and Occupational Therapy) at Miguel Hernandez University in San Juan de Alicante (Alicante, Spain). The majority of students in these faculties are single women, with the following characteristics: interested in studying healthcare sciences, age 18–25 years, and a middle to high socio-economic status.

In Spain, a female college student becoming pregnant without wanting to can be associated with a social stigma. This cultural attitude has existed for many years and still endures in our country, despite the modernization of standards and changes in habits in these situations. FBP may also be related with the situation as perceived by the woman.

### Study design and participants

This cross-sectional observational study, undertaken between February 2005 and February 2009, selected a sample of university students studying healthcare sciences at Miguel Hernández University. The sample comprised all second-year female students studying Occupational Therapy and third-year female students studying Medicine who attended lectures on a particular day during the study period (decided by the research team) and who were willing to participate in the study and complete the questionnaire voluntarily (Table S1). This questionnaire was completed in the lecture halls generally used by the students for their respective degree courses. The inclusion criteria required that the women had to have had vaginal intercourse with a man during the previous three months and have generally used either the male condom or hormonal methods for contraception or not used any contraceptive method at all. This information was assessed by specific questions on the above-mentioned questionnaire (Table S1: General items and Changes in sex life items).
## Variables and measurements

The information was collected with an original questionnaire (Table S1) that was designed by an expert committee, based on their professional experience and relevant scientific articles. Prior to completing the questionnaire it was explained to the students, who were asked to be sincere in their answers. The co-ordination and explanation of the questionnaire were always done by the same researcher, who also resolved any doubts during the process of completion of the questionnaire, which took about 25 min to complete. Previously, a pilot study had been undertaken to assess its comprehension, obtain preliminary data and determine whether the psychometric properties of the questionnaire were adequate. The results of this pilot study showed that the characteristics of the questionnaire were good enough for its future use (analysis of items and internal consistency, indexes of discrimination and factorial analysis) (*Van-der Hofstadt et al., 2007–2008*; *Navarro-Cremades et al., 2013*).

The whole questionnaire collects data about various aspects of female sexual behaviour (Table S1), but in this study we only used those data that the research team considered could be more influential in so far as they provided information about the fear of having an unwanted pregnancy.

The main outcome variable of the study was FBP. This was defined as women who maintained sexual relations with a man and who wished to eliminate the FBP, assessed with the question "What aspects of your sex life would you change?" and the possible answer "Eliminate the fear of pregnancy." If the woman answered that she wished to eliminate the fear of pregnancy, the variable was considered positive (yes), otherwise it was considered negative (no). As FBP in young women has not been defined in the literature, the definition was considered by the woman herself. This FBP could be due to such factors as pain during childbirth, fear of parenthood, or fear of society's response to pregnancy outside marriage. Nevertheless, our sample was taken from university students (mean age close to 21 years), most of whom do not want to have a baby. Thus, this fear is mainly due to the impossibility of leading the same sort of life as a young woman who is not pregnant, as the young mother would then have to focus her life on the maintenance and care of her child for a substantial amount of time, in addition to being unable to finish her studies, either temporarily or permanently.

To analyse the possible factors related with this FBP, the research team selected from the questionnaire (Table S1) the following items in order to determine their association with the main outcome variable: stable relationship (yes; no), contraceptive method used (oral contraceptive; condom; none), desire to increase the frequency of sexual relations (yes; no), sexual partner does not always ejaculate (yes; no), desire to prolong the partner's time to orgasm (yes; no), frequency of sexual intercourse with partner (6 → 5–7 times/week; 5 → 3–4 times/week; 4 → 1–2 times/week; 3 → 2–3 times/month; 2 → once/month; 1 → Never) and age (in years).

The stable relationship parameter was included in the data analysis because women in a casual relationship would experience more fear of failure of the contraceptive method. They might be worried about parenthood with a man they barely know, or from lack of

support if they choose to terminate the pregnancy. The use of condoms is correlated with occasional partners since in addition to the benefit of contraception a condom serves as a method for protection from sexually transmitted diseases. Regarding the use of variables concerning the partner's orgasm, it was logical to think that they could be associated with FBP. On the other hand, the frequency of sexual intercourse with partners could affect the likelihood of pregnancy and therefore the rate of FBP. Finally, a younger age could be associated with less sexual experience.

## Sample size

A total of 472 women completed the questionnaire voluntarily and fulfilled all the inclusion criteria (having had vaginal intercourse with a man during the previous three months (assessed via the question *Sexual Orientation*) and usual contraceptive method of oral contraceptives or condom, or no method (assessed via the item *Method of contraception you use*)). The aim was to estimate the proportion of women who experienced FBP. Assuming a confidence of 95% and a maximum expected proportion ($p = q = 0.5$), the expected error in the estimation was 4.5%.

## Statistical analysis

As the study was undertaken in different academic years, we first checked the homogeneity of the courses using the Pearson Chi square or Fisher test (qualitative data), and ANOVA or Kruskal-Wallis (quantitative data). In the event of a difference over time being found the results would be stratified by academic course.

The descriptive analysis was done using absolute and relative frequencies for the qualitative variables and means plus standard deviations for the quantitative variables. A multivariate logistic regression model was used to estimate the adjusted odds ratios (ORs) in order to analyse the association between FBP and the other variables. This model was stratified by being in a stable relationship to minimize the confounding effect of this variable. The ORs were adjusted for contraceptive method used, desire to increase the frequency of sexual relations, frequency of sexual intercourse with partner, partner's ejaculation (satisfactory ability and time) and age (quantitatively). Finally, the multivariate model was used to calculate the prognostic probabilities of FBP, transformed into charts to simplify the interpretation of the results. The likelihood ratio test was carried out for the goodness-of-fit of the model. All the analyses were done with a Type I error of 5% and the confidence intervals (CI) were calculated for the more relevant study parameters. The analyses were calculated using IBM SPSS Statistics 19.0.

## Ethical considerations

The Ethics Committee of Miguel Hernández University, Elche, (Project Evaluation Organism) evaluated and authorized the study (reference DMC.FNC.01.14), ensuring the voluntary, anonymous free and agreed participation, with no reward or punishment concerning their university studies. In addition, the confidentiality of all the data obtained from the questionnaires was guaranteed. Finally, the participants were informed verbally about the study and about the information required.

**Table 1  Analysis of being in a stable relationship among female university healthcare students in a Spanish region. 2005–2009 data.** Frequency of sexual intercourse with partner (6 → 5–7 times/week; 5 → 3–4 times/week; 4 → 1–2 times/week; 3 → 2–3 times/month; 2 → once/month; 1 → Never).

| Variable | Total<br>$n = 472\ n(\%)/x \pm s$ | In a stable relationship<br>$n = 293(62.1\%)\ n(\%)/x \pm s$ | Not in a stable relationship<br>$n = 179(37.9\%)\ n(\%)/x \pm s$ | p-value |
|---|---|---|---|---|
| FBP | 171(36.2) | 104(35.5) | 67(37.4) | 0.671 |
| Contraceptive method: | | | | |
| Oral contraceptive | 91(19.3) | 77(26.3) | 14(7.8) | <0.001 |
| Condom | 333(70.6) | 200(68.3) | 133(74.3) | |
| None | 48(10.2) | 16(5.5) | 32(17.9) | |
| Desire to increase partner's ability to delay orgasm | 208(44.1) | 55(18.9) | 34(19.0) | 0.980 |
| Partner cannot always ejaculate | 20(4.2) | 11(3.8) | 9(5.0) | 0.505 |
| Desire to increase the frequency of sexual intercourse | 89(18.9) | 118(40.3) | 90(50.6) | 0.029 |
| Age (years) | 20.7 ± 2.4 | 20.7 ± 2.6 | 20.7 ± 2.1 | 0.922 |
| Frequency of sexual intercourse with the partner | 3.3 ± 1.2 | 3.1 ± 1.0 | 3.8 ± 1.3 | <0.001 |

**Notes.**

n(%), absolute frequency (relative frequency); x ± s, mean ± standard deviation; FBP, fear of becoming pregnant.

## RESULTS

A total of 601 female students attended lectures on the days the research team decided to give out the questionnaire. Of these, nine students declined the invitation to participate and left the lecture hall whilst the remainder completed the questionnaire. A total of 15 women had had no sexual activity during the previous three months and were therefore excluded, 7 students failed to hand in the questionnaire and 5 left it completely blank; thus, a total of 565 questionnaires were completed. Of these, and for this particular study, we also excluded another 93 women who used a different contraceptive method to that analysed in this study. This gave a final sample of 472 participants who commonly used as contraception either the condom or oral contraceptives, or who used no method of contraception.

No differences in any of the study variables (or any other variable) were found between the academic years, as all the p-values were above the threshold established (5%).

In the final sample 171 women experienced FBP, equivalent to 36.2% of the total sample (95% CI [31.9–40.6%]). Table 1 shows the descriptive characteristics of the study sample, the mean age of which was almost 21 years (20.7). The contraceptive method most commonly used was the condom (70.6%), followed by oral contraceptives (19.3%). About half the women desired to have a partner with a greater ability to delay orgasm (44.1%) and most of the partners ejaculated in all their relations (95.8%). In addition, 18.9% of the women desired to increase the frequency of their sexual relations and 293 women (62.1%) were in a stable relationship. Comparison of the main characteristics taking into account whether the woman was in a stable relationship showed significant differences ($p < 0.05$) in the following variables: contraceptive method ($p < 0.001$), desire to increase the frequency of sexual intercourse ($p = 0.029$) and frequency of sexual intercourse with the partner

**Table 2  Analysis of fear of becoming pregnant among female university healthcare students in a Spanish region. 2005–2009 data.** Frequency of sexual intercourse with partner (6 → 5–7 times/week; 5 → 3–4 times/week; 4 → 1–2 times/week; 3 → 2–3 times/month; 2 → once/month; 1 → Never). Goodness-of-fit of the models: (1) In a stable relationship: $X^2 = 36.8$, $p < 0.001$; (2) Not in a stable relationship: $X^2 = 15.5$, $p = 0.017$.

| Variable | Adj. OR for women in a stable relationship (95% CI) | p-value | Adj. OR for women not in a stable relationship (95% CI) | p-value |
|---|---|---|---|---|
| Contraceptive method: | | | | |
| Oral contraceptive | 1 | <0.001 | 1 | 0.101 |
| Condom | 3.49 (1.74, 7.02) | | 2.13 (0.53, 8.54) | |
| None | 0.68 (0.13, 3.59) | | 4.56 (0.99, 21.06) | |
| Desire to increase partner's ability to delay orgasm | 1.33 (0.70, 2.53) | 0.393 | 3.59 (1.60, 8.04) | 0.002 |
| Partner cannot always ejaculate | 0.78 (0.18, 3.26) | 0.728 | [a] | [a] |
| Desire to increase the frequency of sexual intercourse | 1.46 (0.86, 2.48) | 0.163 | 1.11 (0.56, 2.21) | 0.770 |
| Age (years) | 0.92 (0.81, 1.04) | 0.180 | 0.87 (0.71, 1.05) | 0.147 |
| Frequency of sexual intercourse with the partner | 1.31 (0.81, 1.04) | 0.047 | 0.98 (0.75, 1.27) | 0.866 |

**Notes.**

FBP, fear of becoming pregnant; Adj. OR, adjusted odds ratio; CI, confidence interval.

[a] This variable was not included in the model due to convergence problems.

($p < 0.001$). However, the rates of FBP were similar between the groups ($p = 0.671$) and these rates had the following values when we differentiated between relationship and contraceptive method: (1) women in a stable relationship (oral, 15.6%; condom, 45.0%; none, 12.5%); (2) women not in a stable relationship (oral, 21.4%; condom, 36.8%; none, 46.9%).

In women in a stable relationship (Table 2), the factors significantly associated with the main outcome variable were type of contraceptive method used (condom) and higher frequency of sexual intercourse with the partner. On the other hand, for women not in a stable relationship, the only significant factor was the desire to increase the partner's ability to delay orgasm, although the contraceptive method (condom or none) had significant values of ORs with a $p$-value close to 5% ($p = 0.101$).

Figure 1 shows the prognostic probabilities of FBP obtained from the multivariate model. Of note in the women in a stable relationship (Fig. 1A) was the great difference between those who commonly used oral contraceptives or no method of contraception versus those who used a condom. Figure 1B shows the increase in the probability of FBP in relation to the less effective contraceptive methods.

## DISCUSSION

### Summary

Our results showed an important proportion of women experiencing FBP, as almost one in three answered the item assessing this aspect affirmatively. Among the factors analysed, use of a condom and higher frequency of sexual intercourse with the partner (only women in a stable relationship), and desire to increase the partner's ability to delay orgasm (only for women not in a stable relationship) were associated with this fear. Finally, the contraceptive
## A. Being in a stable relationship.

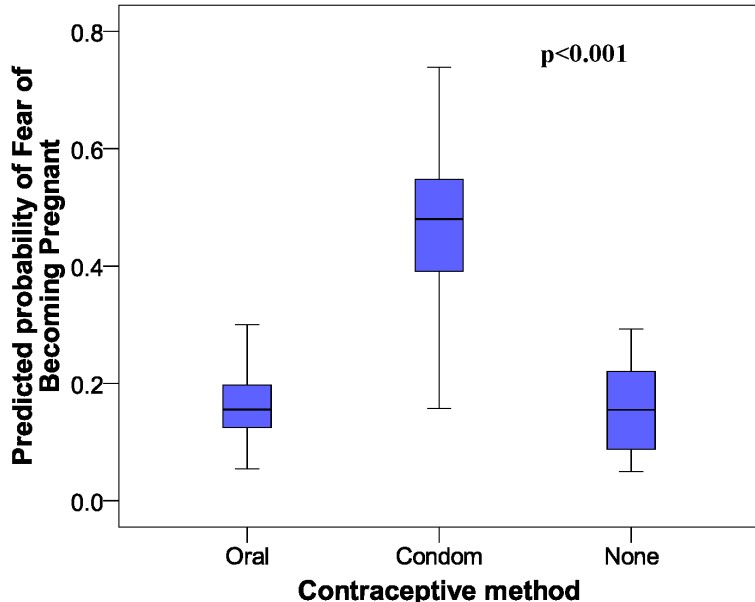

## B. Not being in a stable relationship.

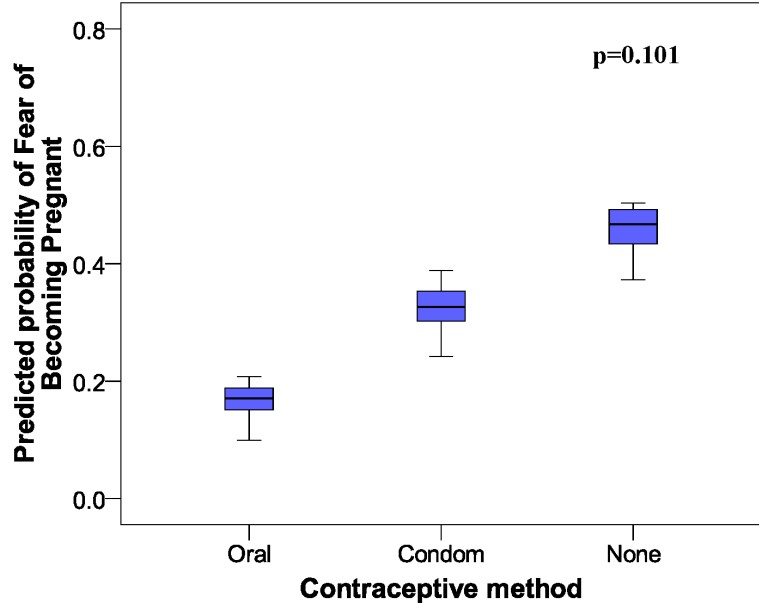

**Figure 1** Probability of fear of becoming pregnant among female university healthcare students in a Spanish region. 2005–2009 data.

method as an associated factor for FBP in women not in a stable relationship was very relevant and close to statistical significance.

## Strengths and limitations of the study

The main strength of this study resides in the lack of other studies analysing the magnitude of this problem and its association with female sexual behaviour. Thus, our results are innovative and indicate the magnitude of FBP among women who use the two most common contraceptive methods or who do not use any contraceptive method. In addition, we also analysed factors associated with sexual practices indicating which groups are more likely to experience FBP.

The study limitations are determined by its design. As it was a cross-sectional study we are unable to determine any temporality between the factors studied and our main variable. This would require future longitudinal studies with predictive models to determine which women are more likely to develop FBP and then intervene early with educational activities to prevent the problem (beliefs about the efficacy of contraceptives and not using any contraceptive method).

We may have committed information bias, as there could be some women who in fact did wish to become pregnant or they may have already had a miscarriage, abortion or live birth, which would justify the fear. However, these would be in the minority, as women in Spain have their first child at the age of 31.6 years and the mean age of our sample was 20.7 years (*Instituto Nacional de Estadística, 2012*). On the other hand, the source of all the information was a questionnaire with good psychometric properties (*Van-der Hofstadt et al., 2007–2008*; *Navarro-Cremades et al., 2013*) that gave reliability to the responses. In addition, we have defined the concept of FBP for young women using their own opinion about the concept; therefore further studies are needed to assess all the aspects of this fear (pain during childbirth, parenthood, society's response to pregnancy outside marriage or impossibility of following the same sort of life as a young woman who is not pregnant).

Finally, to minimize selection bias, the participants selected had all studied sexology in their respective university courses. The students were therefore aware of both the use and the efficacy of the contraceptive methods for preventing an undesired pregnancy or a sexually transmitted disease.

## Comparison with the existing literature

We have only found one study analysing, as a secondary aim, FBP in women who used just hormonal contraceptives (the main aim of the study was to assess noncompliance in their use). This study was a subanalysis evaluating relief on starting the menstrual cycle after having had vaginal sexual intercourse involving male ejaculation, whereas in our study FBP was assessed differently, analysing women who used oral contraceptives, a condom, or no method at all. Thus, we can only compare our results for the women who used oral contraceptives. The magnitude found in the other study was 43.9%, much higher than in our study (15.6% and 21.4%, depending on whether the woman was or was not in a stable relationship). The difference could be related to the fact that our sample comprised persons studying healthcare sciences, and they were thus educated in both gynaecology and

sexually transmitted diseases, whereas the sample in the other study was taken from the general population (*Lete et al., 2008*).

Concerning the factors associated with FBP, we found that the women were more afraid when they used a condom than when they used oral contraceptives. This could be a result of their awareness about the lack of effectiveness of the condom when not used adequately (*World Health Organization, 2011*). Furthermore, when women were not in a stable relationship, they experienced more FBP when they did not use any contraceptive method. Accordingly, it would be advisable to perform qualitative studies to attempt to determine the reasons for the causes of this fear (religious, economical, etc.). On the other hand, for women not in a stable relationship there was a direct relation between male ejaculation and FBP. This could be because, for greater safety, the partners withdrew the penis at the time of ejaculation and on some occasions early ejaculation would not permit this to be done, producing fear in the woman. Finally, a higher frequency of sexual intercourse was logically associated with FBP in women in a stable relationship.

### Implications of the research

This study provides relevant information about understanding contraception among women who will shortly become healthcare professionals. The results show that many of the women experienced FBP depending on whether they used an effective contraceptive method. Furthermore, one in ten students used no contraceptive method at all. We also found an association between early ejaculation/higher frequency of sexual intercourse and FBP. These results could be expected if none of the women used any contraceptive method at all. However, around 90% of the women did use contraception and this should not influence the fear of a possible pregnancy if the method is used correctly.

The results found suggest the need to implement sexual education programmes in the faculties teaching healthcare sciences, aimed at reducing the number of unwanted pregnancies and sexually transmitted diseases, both among these women and among the general population who will in the future be seen by these women. The planning of these programmes would benefit from qualitative studies to help understand the reasons behind this fear, as the responses given would be very useful in designing these educational programmes. Moreover, as well as the aforementioned qualitative studies, other studies could also be conducted to determine whether the proper use of contraceptives results in less FBP; i.e., assessing incompliance taking oral contraceptives and the proper use of male condoms. This could help us determine whether a woman who is aware that she is using contraception correctly experiences less FBP.

### CONCLUSIONS

A high proportion of women soon to become healthcare professionals experienced FBP, depending on the use of efficient contraceptive methods. This may be due to a lack of understanding about the correct use and efficacy of these methods, possibly because of a lack of sexual education in the healthcare sciences faculties. A possible solution to this problem involves educational programmes to help eliminate incorrect beliefs and attitudes about the use of contraceptive methods when the male partner ejaculates, as well

as information about how to use each contraceptive method effectively. Qualitative studies would be useful to design these educational programmes.

## ACKNOWLEDGEMENTS

The authors thank the Department of Applied Psychology at Miguel Hernández University, Elche, for allowing us to use the questionnaire for this study, and Felipe Navarro Sánchez for helping with the computerised database. The authors also thank Maria Repice and Ian Johnstone for help with the English language version of the text.

### Funding

The authors declare there was no external funding for this work.

### Competing Interests

Antonio Palazón-Bru serves as an Academic Editor for PeerJ.

### Author Contributions

- Felipe Navarro-Cremades conceived and designed the experiments, performed the experiments, wrote the paper, reviewed drafts of the paper.
- Antonio Palazón-Bru conceived and designed the experiments, analyzed the data, wrote the paper, prepared figures and/or tables, reviewed drafts of the paper.
- María del Ángel Arroyo-Sebastián, Luis Gómez-Pérez, Armina Sepehri, Salvador Martínez-Pérez, Dolores Marhuenda-Amorós, María Mercedes Rizo-Baeza and Vicente Francisco Gil-Guillén conceived and designed the experiments, reviewed drafts of the paper.

### Human Ethics

The following information was supplied relating to ethical approvals (i.e., approving body and any reference numbers):

The Ethics Committee of Miguel Hernández University, Elche, (Project Evaluation Organism) evaluated and authorized the study (reference DMC.FNC.01.14), ensuring the voluntary, anonymous free and agreed participation, with no reward or punishment concerning their university studies. In addition, the confidentiality of all the data obtained from the questionnaires was guaranteed. Finally, the participants were informed verbally about the study and about the information required.

### Data Availability

The following information was supplied regarding the availability of data:

All data associated with this study are contained in the Supplemental Information.

### Supplemental Information

Supplemental information for this article can be found online at http://dx.doi.org/10.7717/peerj.1200#supplemental-information.

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
