# Peer review of "Fear of becoming pregnant among female healthcare students in Spain"

_PeerJ, doi:10.7717/peerj.1200_

## Round 0.1 · original submission · Major Revisions

Dear Authors, the manuscript has been reviewed by two reviewers that have raised major concerns that need to be addressed prior to any further consideration for publication.

Reviewer 1 ·

Basic reporting

This paper explores an insufficiently studied topic in human sexual behavior, the fear of becoming pregnant (FBP). The authors investigate the possible factors related with FBP. They analyze the association between different contraceptive methods, the desire to increase the frequency of sexual relations and sexual partner ejaculation with FBP through analysis of questionnaires. Overall, this is an interesting study expanding our knowledge about thoughts and emotions in relation to contraceptive methods.
However, there are several methodological issues which make some of the final conclusions questionable and unclear.

Experimental design

There are fundamental flaws in the experimental design. To start, it is not clear what the objective of this study is. If the objective was to identify a cause for FBP then the interpretation of the results is problematic since factors that can be correlated with the main outcome are neglected. In the questionnaire, participants where asked whether they were in a stable relationship. This parameter was not included in the data analysis. This might be a confounder because women in an occasional relationship would fear more from failure of contraceptive method. They might be worried from parenthood with a man they barely know, or from lack of support in case they choose to terminate the pregnancy. The use of condoms is correlated with occasional partners since in addition to the benefit of contraception a condom serves as a method for protection from sexually transmitted diseases. To conclude, women involved in occasional relationships would be more likely to use condoms and would be more likely to suffer from FBP, making the authors interpretations problematic. This problem can perhaps be addressed by including more variables in the data analysis in order to avoid this confounding effect.

Regarding data gathering, the main outcome was defined only as a subjective definition (the subject answering positive to the question, "To eliminate the fear of pregnancy” to question 16). Fear of Becoming Pregnant (FBP) is not a common term in the literature or in common language. Therefore, this term might have been misinterpreted when replying to question 16 (e.g. fear of pain during childbirth, fear of parenthood, fear of society's response to pregnancy outside of marriage etc.). The authors should have given a clear definition to FBP, and the study would have been more coherent if the questionnaire referred to different aspects of FBP.

Validity of the findings

The main finding of this article is that women using condoms experience more FBP than the ones using oral contraceptives. It seems that the findings do not sufficiently support the conclusions. The authors conclude that the positive correlation between the use of condoms and FBP may be explained by: "lack of confidence in the effectiveness of the condom to prevent an undesired pregnancy". Not only this conclusion ignores the confounding effect of other relevant variables, but it is an attempt to draw causation from correlation. It might be that women that tend to fear of becoming pregnant more than others choose to use condoms from other reasons. If the other way around (i.e. using condoms causes FBP) makes more sense than the above, it should be reasoned.

Furthermore, the authors conclude that “The results are worrying, as many of the women experienced FBP independently of whether they used an effective contraceptive method.” (line 220). Contraceptive method effectiveness is cardinally affected by the way one uses it. Women might be experiencing FBP since they are aware of their inconsistent and incorrect use. In that case, this would not be “worrying” results, but rather encouraging. The subjects should have been asked not only which method they use but also regarding their adherence to correct use.

Additional comments

English needs tightening in places and the manuscript would benefit from a proof-read by a native speaker.

·

Basic reporting

Pregnancy in unwed women is considered taboo in some society and more accepted in others. The article can benefit by adding information regarding the Spanish society general attitude to this phenomenon. This information can be add to the introduction section.
In future studies it will be interesting to know the reason for the fear of becoming pregnant each women (religious, economical and etc.)

Experimental design

contraception method, desire to increase the frequency of their sexual relation, whether the partner always ejaculate, desire to prolong time to orgasm and age were examined for their association with the outcome variable, why did the authors expect this variables to influence the results?

Why was the number of intercourse in the last three month were not asked (the authors settled for one or more) as it likely to affect the likelihood of pregnancy and therefore the rate of fear of becoming pregnant.

Validity of the findings

No Comments

Additional comments

The article contain no information regarding the women pregnancy history (miscarriages, abortions and live birth) which may affect the results.

---

## Round 0.2 · Major Revisions

The manuscript has improved but I would like to turn the authors to the first comment of the reviewer :"*One of the major issues raised in the previous review was the need to minimize the effect of confounding factors such as 'stability of the relationship'. This issue was partially and insufficiently addressed. The authors added that “293 women (62.1%) were in a stable relationship“, but failed to note how many of the “condom group” were in a stable relationship and how many of the “oral contraceptive group” were in a stable relationship. Without comparison of the two groups the confounding effect cannot be ruled out. "
Addressing this comment satisfactorily is essential for the acceptance of this manuscript for publication

Reviewer 1 ·

Basic reporting

Many of the comments were adequately addressed, however some points need further revision.

Experimental design

*One of the major issues raised in the previous review was the need to minimize the effect of confounding factors such as 'stability of the relationship'. This issue was partially and insufficiently addressed. The authors added that “293 women (62.1%) were in a stable relationship“, but failed to note how many of the “condom group” were in a stable relationship and how many of the “oral contraceptive group” were in a stable relationship. Without comparison of the two groups the confounding effect cannot be ruled out.
*In regard to my comment on the definition of FBP, the authors described what the participants might have had in mind when they positively answered that they wished 'to eliminate the fear of pregnancy': "The definition of FBP was that considered by the woman herself. This could be due to such factors as pain during childbirth, fear of parenthood, or fear of society's response to pregnancy outside marriage". However, I think that the authors should include a clear scientific definition of this unfamiliar term, supported by literature. If this term was never adequately defined then it should be clearly defined by the authors.

Validity of the findings

*I have previously noted that the use of judgmental expression “The results are worrying" is inappropriate as appeared in the original text. I find the revised sentence ("The results are encouraging, as many…") problematic in the same manner.

---

## Round 0.3 · accepted · Accept

The authors have well addressed all the comments of the reviewers and this paper can be accepted for publication in its current format